# The Emulsification and Stabilization Mechanism of an Oil-in-Water Emulsion Constructed from Tremella Polysaccharide and Citrus Pectin

**DOI:** 10.3390/foods13101545

**Published:** 2024-05-16

**Authors:** Fangwei Liu, Weiwei He, Xiaojun Huang, Junyi Yin, Shaoping Nie

**Affiliations:** State Key Laboratory of Food Science and Resources, China-Canada Joint Laboratory of Food Science and Technology (Nanchang), Key Laboratory of Bioactive Polysaccharides of Jiangxi Province, Nanchang University, Nanchang 330047, Chinaweiweihe_fs@ncu.edu.cn (W.H.); huangxiaojun0617@163.com (X.H.); yinjy@ncu.edu.cn (J.Y.)

**Keywords:** emulsions, tremella polysaccharide, pectin

## Abstract

The objective of this study was to investigate the feasibility of the mixture of tremella polysaccharide (TP) and citrus pectin (CP) as an emulsifier by evaluating its emulsifying ability/stability. The results showed that the TP:CP ratio of 5:5 (*w*/*w*) could effectively act as an emulsifier. CP, owing its lower molecular weight and highly methyl esterification, facilitated the emulsification of oil droplets, thereby promoting the dispersion of droplets. Meanwhile, the presence of TP enhanced the viscosity of emulsion system and increased the electrostatic interactions and steric hindrance, therefore hindering the migration of emulsion droplets, reducing emulsion droplets coalesce, and enhancing emulsion stability. The emulsification and stabilization performances were influenced by the molecular weight, esterified carboxyl groups content, and electric charge of TP and CP, and the potential mechanism involved their impact on the buoyant force of droplet size, viscosity, and steric hindrance of emulsion system. The emulsions stabilized by TP-CP exhibited robust environmental tolerance, but demonstrated sensitivity to Ca^2+^. Conclusively, the study demonstrated the potential application of the mixture of TP and CP as a natural polysaccharide emulsifier.

## 1. Introduction

Emulsification is a pivotal process in the food processing industry, which involves the mixing of the oil–water phase system in the presence of an emulsifier to create a stable and uniform system [1,2]. Polysaccharides are the preferred emulsifier due to their excellent emulsifying stability [3] and their ability to reduce the dosage of synthetic small molecule emulsifiers [4,5]. Currently, various polysaccharides are applied in emulsion products, including xanthan gum, carrageenan, modified starch and carboxymethyl cellulose [5]. With rising consumer expectations for food quality, there is increasing interest in the functionality of food and plant-based ingredients.

*Tremella fusiform* Berk is an edible fungus that is rich in nutrients, including dietary fiber, protein, vitamins, and a variety of amino acids. Tremella polysaccharide (TP) is considered to be the key ingredient of tremella, with potential effects on human nutrition and health, including immunoregulation, anti-tumor, anti-oxidation, glycemia and lipid regulation, and anticoagulation [6]. TP is a heteropolysaccharide with a backbone consisting of α-(1→3)-D-mannose and branched chains composed of glucose, mannose, fucose, xylose, and glucuronic acid [7,8]. Research has demonstrated that the emulsifying properties of TP depend on its molecular characteristics, including molecular weight, monosaccharide composition, and protein content [9]. Polysaccharides with a high protein content and low molecular weight exhibit emulsifying activity well, while those with a high molecular weight demonstrated superior emulsification stability [10]. Additionally, electrostatic repulsion and viscosity play a crucial role in maintaining the stability of the emulsion system [7].

Pectin, known for its complex structure in the polysaccharide family [11], exhibits various bioactive effects, including improving intestinal health, absorbing heavy metal ions, anti-tumor, and hypoglycemia [12,13,14]. Citrus pectin (CP) is composed primarily of a backbone of galacturonic acid, and some hairy regions consisting of rhamnose, galactose, and glucose [15]. Research has demonstrated that the hairy regions of pectin act as a steric barrier, while the ionization of carboxylic groups (-COOH) results in a negative charge on the droplets, leading to electrostatic stabilization [3]. Additionally, the total electrical charge and its distribution pattern within the pectin molecule, which is determined by its degree of esterification, significantly impact the composition, thickness, and configuration of the adsorbed interfacial film [16].

TP and CP each exhibit unique advantages in emulsion stability and emulsification, respectively. In addition, the development of an emulsifying agent based on TP and CP can meet the consumers’ demands for food ingredients. Therefore, the purpose of the present study was to investigate the feasibility of the mixture of TP and CP as an emulsifier. The behavior of TP and CP in the emulsions was evaluated by multiple light scattering and optical microscope. Fluorescence microscope, ζ-potentials, and rheology were used to explain the emulsification and stabilization mechanism of polysaccharides. Additionally, the stability of emulsions to environmental stress (pH and ionic strength) was investigated to evaluate its potential as an emulsion stabilizer used in the food industry.

## 2. Materials and Methods

### 2.1. Materials and Reagents

*Tremella fuciformis* Berk was purchased from Ningde, Fujian, China. Pectin (CAS: 9000-69-5, Galacturonic acid ≥ 58.0%, Methoxyl Group ≥ 6.0%) was purchased from Solarbio Life Sciences Co., Ltd., Beijing, China. Corn oil was obtained at a local supermarket in Nanchang, Jiangxi, China. The stain (Nile Red and fluorescein isothiocyanate) and papain were supplied by Sigma-Aldrich Co., Ltd., Shanghai, China. The thermostable α-amylase was obtained from Aladdin Co., Ltd., Beijing, China. Other chemicals were analytical grade.

### 2.2. Extraction of Polysaccharide from Tremella fuciformis Berk

TP was extracted from the fruiting bodies of *Tremella fuciformis* Berk, after the pre-processing of milling and sieving (200-mesh). The TP was extracted according to the protocol [17] with some modifications. Shortly, the powder was added into the hot water (stock rate: 1:100 *w*/*v*, 95 °C, 2 h) and stirred for extracting water-soluble polysaccharide. After the centrifugation (12,000 rpm, 15 min, 25 °C), the supernatant was collected and the precipitate was added into the hot water (stock rate: 1:50 *w*/*v*, 95 °C, 2 h) again. The thermostable α-amylase and papain (at a raw material of 0.5%, 60 °C, 30 min, Ph = 6) were added into the supernatant in sequence for removing starch and protein, respectively. After enzyme inactivation (100 °C, 20 min) and centrifugation (12,000 rpm, 15 min, 4 °C), the supernatant was precipitated by ethanol, dialyzed, concentrated and finally lyophilized.

### 2.3. Characterization of Polysaccharides

Protein content of polysaccharide was analyzed by the Coomassie Bright Blue method, with serum albumin as the standard [18]. The contents of neutral sugar of polysaccharides were determined by the phenol-sulfuric acid colorimetric method, using glucose as the standards. The contents of uronic acid of TP and CP were determined by the sulfuric acid-carbazole method, using glucuronic and galacturonic acid as the standards, respectively [19,20]. The experiments were repeated three times, and the data were expressed as mean ± standard deviation.

The molecular weight of polysaccharides was detected by high performance gel permeation chromatography (HPGPC) equipped with 2414 RI detector and Ultrahydrogel^TM^ linear gel column (7.8 × 300 mm, Waters, Milford, MA, USA). The samples were dissolved in mobile phase (0.1 M NaNO_3_ solution) and passed through a 0.22 μm water phase membrane filter [21].

The monosaccharide compositions of polysaccharides were detected by Dionex ICS-6000 ion exchange chromatography system. The samples were prepared as follows: 5 mg polysaccharide were hydrolyzed with 0.5 mL 12 M H_2_SO_4_ for 30 min under an ice-bath condition. After being diluted with 2.5 mL ultra-pure water, the samples were further hydrolyzed in an oil-bath condition at 100 °C for 2 h. The hydrolyzed samples were passed through a 0.22 μm water phase membrane filter, after diluting a certain volume. Nine monosaccharides, including fucose (Fuc), arabinose (Ara), rhamnose (Rha), galactose (Gal), glucose (Glc), xylose (Xyl), mannose (Man), galacturonic acid (GalA), glucuronic acid (GlcA), had been used as standard samples. The experiments were repeated three times [21].

The degree of methyl-esterification (DM) and acetylation (DA) of pectin was determined by the Bruker AVANCE III HD 400 MHz NMR spectrometer [22], after the pectin were deuterium-exchanged by lyophilization from D_2_O (5:1 *w*/*v*) for three times, with 0.1 mL 0.2 mg/mL sodium 3-(trimethylsilyl)-propionate-D4 worked as the internal standard.

The polysaccharides were lyophilized and measured using Fourier transform infrared (FT-IR, Thermo Fisher Co., Ltd., Waltham, MA, USA) with a spectral of 400–4000 cm^−1^ [23]. Samples were tableted with KBr powder for spectra collections.

### 2.4. Emulsion Preparation

Emulsion preparation referred to the method reported by Liu et al. [24] with some modifications. According to the preliminary experimental results, it was found that emulsion containing a 1.0% polysaccharide solution exhibited more stable emulsifying properties and consumed less polysaccharide compared to emulsion with other concentrations of polysaccharide. Besides, 1.0% TP:CP = 5:5 had better emulsifying stability with 1.5 mL oil than using one polysaccharide alone. Thus, the emulsion preparation method in this study was performed as follows: the 6 mL of the polysaccharide solution (1.0%, *w*/*v*) was mixed with 1.5 mL corn oil by a high-speed disperser (XHF-DY, Ningbo science biotechnology Co., Ltd., Ningbo, China) at 11,000 rpm for 120 s. The polysaccharide solutions were prepared by component of TP and CP in different proportions, including TP:CP = 10:0, TP:CP = 7:3, TP:CP = 5:5, TP:CP = 3:7, TP:CP = 0:10 (*w*/*w*). Braveds MB-1 liquid preservative (0.1% of total volume) had been used for preventing microbial growth.

### 2.5. Multiple Light Scattering

The stability and size distribution of emulsion were evaluated by TURBISCAN TOWER (Formulaction Smart Scientific Analysis Co., Ltd., Toulouse, France) applying the principle of the multiple light scattering. The scanning time interval for testing was set at 2 h [1]. The samples were stored at 4 °C in the refrigerator during the non-detection period, while the determination was performed at 25 °C to accelerate the emulsification instability. The refractive index of disperse phase and aqueous phase were 1.477 and 1.333, respectively.

### 2.6. Emulsion Microstructure

Optical microscope. The microstructure of emulsions was observed by optical microscopy (Olympus CKK53) at room temperature (25 °C). A total of 10 μL of emulsion was placed on the microscope slide and carefully covered. The photomicrographs were taken under 100× magnifications [5].

Fluorescence microscope. The fluorescence microscope (Leica Co., Ltd., Wetzlar, Germany) was used to observe polysaccharide and corn oil in the emulsions, as well as the localization of polysaccharide at the oil–water interface. A total of 100 μL of fresh emulsion was stained with 20 μL mixed dye solution containing 0.1 wt% Nile red and 0.1 wt% fluorescein isothiocyanate (FITC). A small drop of emulsion was placed on the microscope slide and carefully covered [25].

### 2.7. ζ-Potential Analysis

The ζ-potential of freshly prepared emulsions were determined by a Nanosize and Zeta potential analyzer (Malvern Instrume3nts, Worcestershire, UK) after being diluted 50 times [7,26]. The experiments were repeated three times.

### 2.8. Rheology

The apparent viscosity of samples was determined in a shear rate range from 0.01 to 1000 s^−1^ at 25 °C using an ARES-G2 rheometer (TA instruments Co., Ltd., New Castle, DE, USA), equipped with a parallel plate geometry (40 mm in diameter), with a gap distance of 0.5 mm [21]. The samples included the 1.0% TP-CP solutions with different proportion and the 1.0% TP-CP = 5:5 solution with NaCl or urea [27]. The experiments were repeated three times.

### 2.9. Statistical Analysis

The statistical differences were evaluated by analysis of variance (ANOVA) and Tukey test at 5% significance using SPSS 27.0 software (IBM SPSS, Chicago, IL, USA). Origin software (Version 8.0) was used to plot the experimental data.

## 3. Results and Discussion

### 3.1. The Physicochemical Properties of the Polysaccharides

The physicochemical properties of the polysaccharides are shown in Table 1. TP had a molecular weight of 2720 kDa, and was composed of a large amount of Man, with a small of Fuc, Xyl, and GlcA, and a trace of Glc. Consistent with previous studies, TP was mainly composed by a backbone of mannose and some hairy regions consisted with glucose, mannose, fucose, xylose, and glucuronic acid [7,8,28]. CP had a molecular weight of 1280 kDa, and was composed of a large amount of GalA, with a small of Gal and Rha, and a trace of Glc. DM and DA represent the amount of methoxy and acetyl groups capable of replacing the carboxylic acid groups in GalA residues, respectively [29]. In this research, the amount of DM and DA of CP was 69.38% and 2.02%, respectively, which could be categorized as pectin with high degree of methoxylation. The previous literature on citrus pectin has reported that DM and DA were detected in the range of 66.33%~92.00% and 0.03%~3.16%, respectively [29].

FT-IR spectra could be used to describe the functional groups and bonding information of polymers, and the FT-IR spectra of those polysaccharides is shown in Figure 1. In our study, the absorption peaks between 3200 to 3600 cm^−1^ were attributed to intramolecular and intermolecular O-H stretching vibrations. TP and CP had absorption peaks at 1740 cm^−1^, indicating that the presence of esterified carboxyl groups. Generally, the esterified carboxyl group in polysaccharide can adsorb onto the oil-and-water interface [16], which is an important parameter in assessing the emulsifying activity of polysaccharides [27,30]. As shown in the FT-IR spectra, TP had the lesser intense peaks at band area (1740 cm^−1^) of esterified carboxyl groups than those of CP, which may be related to the weaker emulsifying activity of TP [31]. The observed absorption bands at 1418 cm^−1^ and 1617 cm^−1^ can be assigned to asymmetric and symmetric stretching of the free carboxylate group, respectively [9,32]. No new absorption peaks were observed in the mixture TP-CP compared with TP and CP alone, suggesting that no groups formed between TP and CP; that is, no covalent binding formed between them [33].

### 3.2. Performance of Different Systems on Emulsifying Ability

The emulsifying ability refers to the capacity to create an interfacial film, essentially the ability to form emulsified oil drops. In this work, it was demonstrated that TP, CP, and their mixture possessed emulsifying activity, as shown in Figure 2A. The average droplet size and emulsion morphologies are illustrated in Figure 2B and Figure 2C, respectively. The average droplet sizes of emulsion systems (TP:CP = 10:0, TP:CP = 7:3, TP:CP = 5:5, TP:CP = 3:7, TP:CP = 0:10) were 31.99 µm, 22.65 µm, 12.62 µm, 8.75 µm, and 7.55 µm, respectively, which were consistent with the results of emulsion droplet morphology. It indicated that the higher content of CP was more favorite for emulsifying ability, which may be attributed to the lower molecular weight and high methyl esterification of CP. It has been reported that polysaccharides with lower molecular weight exhibited a greater tendency to distribute at the oil–water interface, enabling more hydrophobic groups to absorb on the interface, which favored the formation of smaller average droplet size [1,2].

### 3.3. Performance of the Different System on Emulsion Stability

Emulsion instability encompasses phenomena such as droplet flocculation and aggregation, as well as droplet creaming and sedimentation, ultimately leading to a phase separation [34]. To evaluate emulsion stability, the appearance of various emulsions was observed. As shown in Figure 3A, the obvious phase separation developed in the emulsion system (TP:CP = 10:0, TP:CP = 3:7 and TP:CP = 0:10) as the storge time lengthened. Notably, the emulsion system (TP:CP = 7:3 and TP:CP = 5:5) remained stable even up to the 21st day after preparation.

Furthermore, the TURIBISCAN Lab was used to detect particle migration phenomena, as shown in Figure 3B. The back scattering light intensity (BS) is proportional to the dispersed phase volume fraction and inversely proportional to the dispersed phase droplet size [35,36]. The BS obtained by the first scanning decreased with the increase in the TP proportion, indicating that the emulsion with high TP proportion had larger droplet size, as observed in Figure 2B. As to the emulsion stabilized by TP:CP = 3:7, the vertical line of BS translated to the right with time in the measurement cell height range of 0 mm to 24 mm, and the horizontal line of BS was decreased as time increased. This indicated that the larger particles rise at the uniform rate, while the smaller droplets left behind continue to flocculate or coalesce. In the range of 24 mm to 40 mm, horizontal line of BS increased with time, which was caused by the increase in the disperse phase fraction due to the rise of droplets from the bottom.

As to the emulsions stabilized by TP:CP = 10:0 and TP:CP = 0:10, the progression of creaming was gradual with no sharp boundaries. The BS of emulsions (TP:CP = 10:0) had reduced in the range of 0 mm to 30 mm cell height, suggesting some flocculation increasing the size of droplets [37], but the viscosity of solution was high enough to slow down the mass creaming normally, which will be discussed in detail below. Additionally, the emulsions (TP:CP = 0:10) had smaller droplet size with lower buoyant force acting on them, resulting in slowing down the mass creaming. The BS of emulsions (TP:CP = 7:3 and TP:CP = 5:5) remained nearly constant as the duration increased.

### 3.4. Effects of Concentrations, Acid-Alkaline and Ionic Strength on Emulsion Stability

To evaluate the stability of emulsions under varying concentrations, pH levels, and ionic environment, the appearance of emulsions with TP:CP = 5:5 at different storage time (1–21) days were determined, as Figure 4 showed.

Concentration. With the polysaccharide concentration decreased, the emulsions appeared obviously cream layer (Figure 4), corresponding to the increased droplets size (Appendix A). It may be attributed to bridging flocculation, which occurs when the amount of polysaccharide present is insufficient to completely coat the droplet surfaces, resulting in a polysaccharide molecule to adsorb onto the surfaces of several droplets [38].

Ionic strength. The strength of salt ions can affect the colloidal behavior of the colloidal suspension, and ultimately affecting the performance of the emulsions [39]. The emulsion at a low ionic strength (0.5 M Na^+^) appeared an obvious cream layer after 7 days. It was reported that Na^+^ can reduce the electrostatic repulsion intramolecularly and form a tighter conformation of the molecule, which hindered the adsorption at the water/oil interface, leading to instability of the emulsion [25,40]. As Appendix A showed, the emulsion droplets were relatively bigger at a low Na^+^ strength, which promoted the oil droplets coalescence and decreased the emulsion stability. When the emulsion systems were exposed to a high ionic strength (>0.5 M Na^+^), the droplet size of the emulsions decreased, and a small amount of water layer appeared on the first day, but there were no significant changes in the water layer in the following days. This phenomenon could be explained by the fact that sodium ions reduced the electrostatic repulsion among molecules by masking the negative charge on the polysaccharide chains. This promoted intermolecular aggregation, leading to the formation of a polysaccharide polymer layer on the droplet surface and reducing the aggregation between droplets. Additionally, the aggregation of polysaccharide molecules reduced the degree of hydration, resulting in a small amount of water layer in the emulsion [38]. With the Ca^2+^ strength increased, the emulsions appeared obviously cream layer, corresponding with the increased droplets size (Appendix A), which might be caused by salting-out effect [41].

pH value. After a storage time of 21 days, there were no noticeable alternations in the appearance of the emulsions (pH = 8), indicating their resilience to a high alkaline condition. It may be attributed to the electrostatic repulsion in alkaline solution, endowing the emulsions with a higher storage stability [25]. As the storage time increased to 14 days, a discernible cream layer appeared in the emulsion system (pH = 2), which may be associated with the smaller droplet size (Appendix A) of the emulsion. The smaller droplet size lead to a larger oil–water interfacial area and interfacial energy, promoting the coalescence of oil droplets and reducing the emulsifying stability [42,43]. However, as the storage time increased to 21 days, there was a discernible cream layer in the emulsion system (pH = 6), attributing to changes in the electrostatic interactions in the system. The presence of weak acid condition led to a reduction in the surface charge of the droplets, thereby limiting the amount of droplet flocculation [25].

### 3.5. The Effects of Polysaccharides System on Emulsion Properties

#### 3.5.1. Droplet Size and ζ-Potential

Sedimentation caused by the gravity of emulsified oil droplets is known to affect the emulsion stability. The emulsion with smaller droplet size can effectively limit the sedimentation of emulsion due to the reduction in gravitational motion [44]. As shown in Figure 5A, the droplets size of the emulsion decreased with increasing CP proportion. However, it should be noted that the emulsion system (TP:CP = 0:10) was not the most stable one, and the droplets size increased over time. From a thermodynamic perspective, the emulsion with smaller droplet size can lead to a larger oil–water interfacial area and interfacial energy, promoting the coalescence of oil droplets and reducing emulsifying stability [42,43]. It suggested that CP alone was favorable for the formation of smaller droplets, due to the esterified carboxyl groups in CP can adsorb onto the oil-and-water interface to form an interfacial layer, but it may lack sufficient viscosity, steric or electrostatic interactions to maintain the emulsion stability [38]. Similar phenomena were also observed in emulsion system with TP:CP = 3:7.

ζ-potential reflects the contribution of electrostatic repulsion on emulsion stability [45] (Figure 5B). It has been reported that a higher ζ-potential value resulted in increasing electrostatic repulsion between droplets, which improved their stability against aggregation [15]. Consistent with prior findings, the elevated ζ-potential of TP may be ascribed to the abundant presence of free carboxylate groups [9]. The ζ-potential of emulsion system with a TP:CP ratio of 0:10 (−20.93 mV) was significantly lower than other emulsion systems, which supported the above note that the emulsion with CP alone may lack sufficient electrostatic interactions to maintain the emulsion stability [45]. The emulsion stabilized by TP:CP with a ratio of 10:0 exhibited the highest ζ-potential, but it remained unstable. This may be attributed to the larger droplet size in these systems, resulting in higher buoyant force acting on them. Despite the elevated ζ-potential, it was insufficient to counteract this buoyant force.

#### 3.5.2. Rheological Properties

The collision and combination of emulsified oil droplets with each other results in oil–water separation in the emulsion. It is a critical mechanism of emulsifying stability that the collision frequency of emulsified droplets is reduced by increasing the viscosity of emulsion system [46], while the rheological properties of the O/W emulsion mainly depend on the aqueous phase [5,47]. Therefore, the viscosity of polysaccharide solutions (Figure 6) was measured in this work to analyze the mechanism of emulsion stability.

The viscosity of polysaccharide solutions increased with the increasing content of TP, following the order “TP:CP = 10:0” > “TP:CP = 7:3” > “TP:CP = 5:5” > “TP:CP = 3:7” > “TP:CP = 0:10”. Notably, despite the polysaccharide solution of TP:CP = 10:0 having the highest viscosity, it did not exhibit the strongest emulsion stability. This may be attributed to a depletion mechanism, wherein the polysaccharides promoted droplet flocculation in emulsions [48,49]. The depletion flocculation in the emulsion system was mainly caused by the existence of non-absorbing polysaccharide, which induced an osmotic pressure gradient in the aqueous phase surrounding the droplets, and further resulted in the clusters of emulsion droplets [48,49,50,51], as the morphologies of emulsion systems (TP:CP = 10:0) displayed in Figure 2C. The high molecular weight, multi-branch structure and less content of esterified carboxyl groups may result in a low absorbability of TP onto oil droplets [49]. Although the high viscosity can hinder the migration of emulsion droplets, it was insufficient to counteract the buoyant force of droplets, thus merely delaying the process of mass creaming, as Figure 3B (TP:CP = 10:0) showed.

Emulsion with as TP:CP ratio of 5:5 and 7:3 displayed relatively stable resistance to phase separation, despite containing larger particles than the emulsion of TP:CP = 3:7 or TP:CP = 0:10, and having a smaller aqueous phase viscosity than the emulsion of TP:CP = 10:0. This finding is consistent with the previous reports on the emulsions stabilized by xanthan gum [15]. This may be attributed to the dynamic balance among the buoyant force of droplet size, viscosity and electrostatic repulsion in the emulsion system (TP:CP = 7:3 or TP:CP = 5:5). In these system, the viscosity of aqueous phase counteracts the tendency of emulsion droplets to float due to gravity, further inhibiting the macroscopic phase separation [52].

Furthermore, it was found that the droplets size decreased with the decrease in aqueous phase viscosity in Figure 2B and Figure 6. However, the emulsifying ability appeared to remain independent of the aqueous phase viscosity, since a decrease in droplet size was observed as the CP content and viscosity increased in the verification test (Appendix A).

In order to explore the force in the mixing solutions, NaCl (1 M) or urea (6 M) were added to the mixture (TP:CP = 5:5), respectively. A great reduction in the apparent viscosity was observed (Figure 6), suggesting that electrostatic interaction and hydrogen bonding existed in the mixture of TP and CP [53,54]. Additionally, according to the results of HPGPC, polysaccharides (TP:CP = 5:5) did not yield a complex with an elevated molecular weight, as Appendix A showed. Therefore, the present data merely suggested the presence of hydrogen bonding and electrostatic interactions within the blend of TP and CP.

#### 3.5.3. Fluorescence Microscope Analysis of Emulsion System

To gain a visual insight into distribution of polysaccharides and oil droplets within the emulsions, the emulsions stabilized by different TP/CP proportion were analyzed by using fluorescence microscope. As shown in Figure 7, it can be clearly observed that the more oil droplets were wrapped in a layer of green interfacial circles in the emulsion system stabilized by higher content of CP, which reflected the adsorption capacity of CP at the oil–water interface. However, fewer green interfacial circles were observed in the emulsion stabilized by TP alone, suggesting that TP was primarily distributed around the droplets to form the steric hindrance [55].

### 3.6. The Mechanisms of TP-CP on Emulsion Ability and Emulsion Stability

CP played a crucial role in the formation of smaller oil droplets in emulsions, which may be ascribed to its high content of esterified carboxyl groups and smaller molecular weight. However, emulsions stabilized solely by CP trended to exhibit instability, due to the multiple effects of lower aqueous phase viscosity, steric or electrostatic repulsion and high interfacial energy, as exemplified by the emulsion system (TP:CP = 0:10) in Figure 8.

TP can stable the emulsion by providing electrostatic repulsion, slowing down the migration rate of droplets in the aqueous phase, and reducing the probability of collision and coalescence. However, high concentration of TP (with high viscosity) may result in flocculation and further the increasing of droplets size due to the depletion flocculation, even though the aqueous phase with high viscosity can prevent the floccules from creaming, as illustrated in Figure 8.

Therefore, it is speculated that CP, with the low molecular weight and high content of esterified carboxyl groups, trends to promote the dispersion of oil droplets and improve the emulsifying activity. TP, characterized by a high molecular wight, provides higher electrostatic repulsion, viscosity, and steric hindrance to hinder the migration and coalescence of the emulsion droplets, thus improving the stability of the emulsion, as illustrated in Figure 8.

## 4. Conclusions

The results of this study indicated that a TP:CP ratio of 5:5 had a potential application as a novel and natural emulsifier/stabilizer. CP enhanced dispersion of oil, thereby improving its emulsifying ability, whereas the emulsifying stability was weak, since its lower aqueous phase viscosity, and lack of steric or electrostatic repulsion. TP improved the rheology of system, hindered the migration of emulsion droplets, reduced emulsion droplets coalescence, provided electrostatic repulsion and further improved emulsion stability, whereas TP had a low emulsifying ability. The synergistic effects between TP and CP could enhance the stability of emulsion. The mechanism of emulsion stability was associated with the molecular weight, esterified carboxyl groups content, and electric charge of polysaccharides, which affected the droplet size, viscosity, and steric hindrance of emulsion system.

TP and CP exhibit emulsification stability and functional activity, and can serve as substitutes for traditional emulsifiers, thereby meeting consumer demand for food ingredients. In addition, the emulsion stabilized by TP-CP has a specific viscosity and shear thinning behavior, making it a promising option for creating fat-based enteral nutrition products for individuals with dysphagia.

## Figures and Tables

**Figure 1 foods-13-01545-f001:**
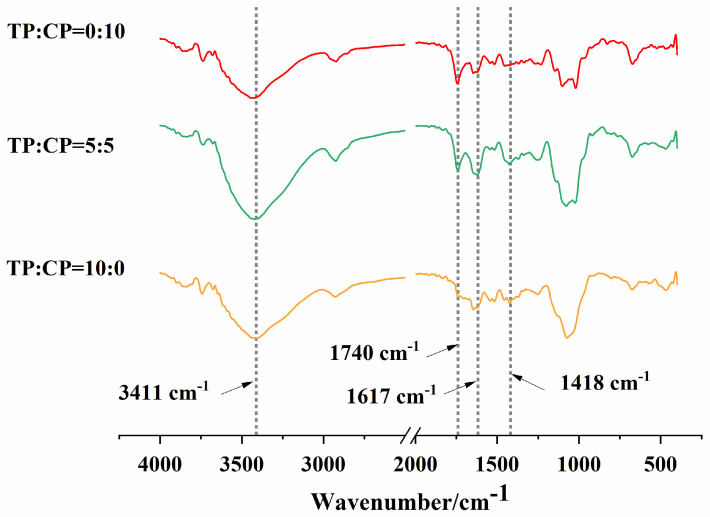
FT-IR spectra of polysaccharides.

**Figure 2 foods-13-01545-f002:**
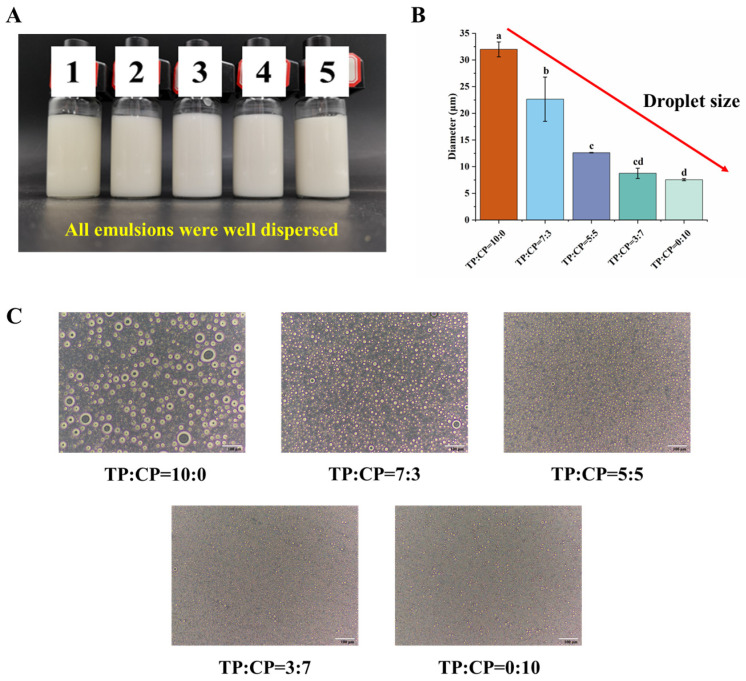
Performance of freshly prepared emulsion on emulsifying ability stabilized by different mixing ratios of 1.0% TP-CP. (**A**) the appearance of emulsions, in which the labels of “1”, “2”, “3”, “4”, and “5”, represented the emulsions stabilized by TP:CP = 10:0, TP:CP = 7:3, TP:CP = 5:5, TP:CP = 3:7, and TP:CP = 0:10, respectively; (**B**) the average droplet size of emulsions; (**C**) the morphologies of emulsions. Different letters in the same chart represent significant differences between different treatments (*p* < 0.05).

**Figure 3 foods-13-01545-f003:**
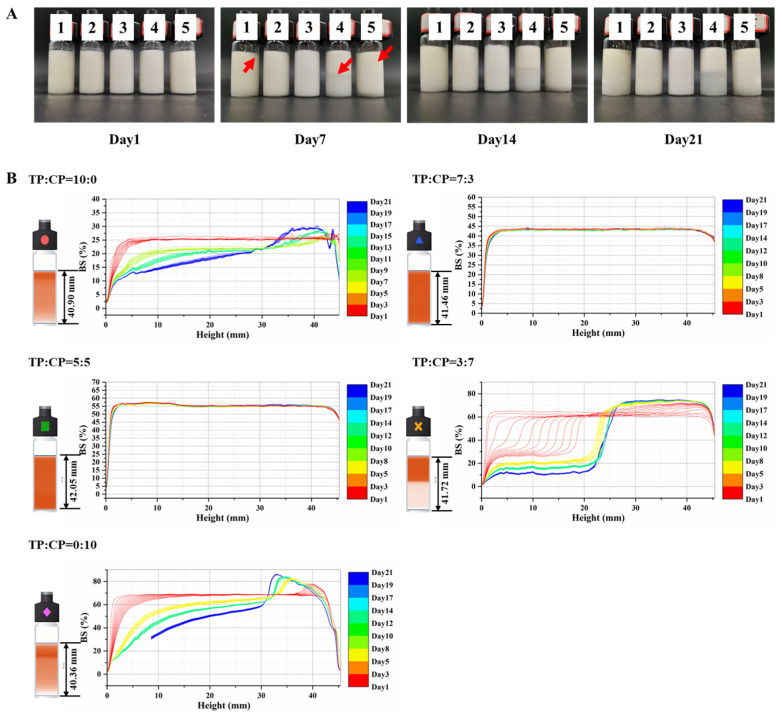
Performance of emulsion systems on stability over 21 days stabilized by different mixing ratios of 1.0% TP-CP. (**A**) the appearance of different emulsion systems, in which the labels of “1”, “2”, “3”, “4”, and “5”, represented the emulsions stabilized by TP:CP = 10:0, TP:CP = 7:3, TP:CP = 5:5, TP:CP = 3:7, and TP:CP = 0:10, respectively. The red arrows represented the initial appearance of cream layer in the emulsions. (**B**) Backscattering profiles of different emulsion systems.

**Figure 4 foods-13-01545-f004:**
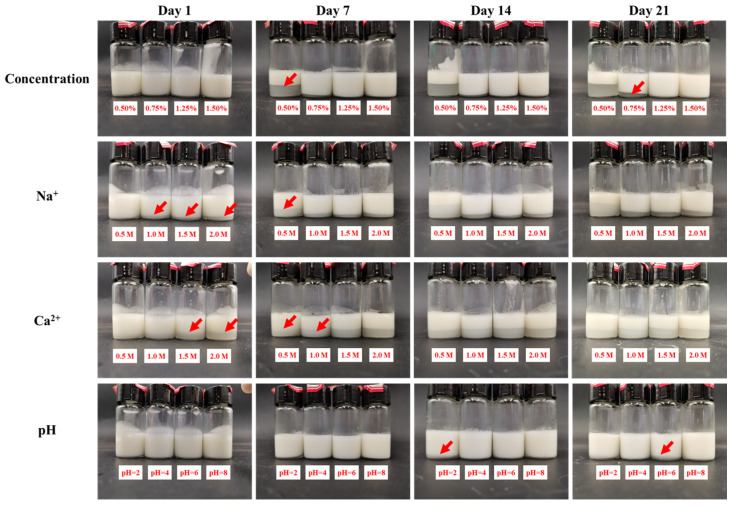
The appearance of emulsion stabilized by TP:CP = 5:5 under different polysaccharide concentration (0.5%, 0.75%, 1.25%, 1.5%), ionic strength (Na^+^ or Ca^2+^: 0.5 M, 1.0 M, 1.5 M, 2.0 M; stabilized by 1.0% polysaccharide) and pH levels (pH = 2, 4, 6, 8; stabilized by 1.0% polysaccharide) after different storage days (1–21 days) at 4 °C. The red arrows represented the initial appearance of cream layer in the emulsions.

**Figure 5 foods-13-01545-f005:**
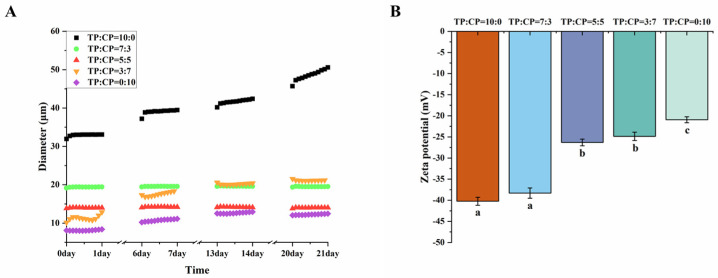
Properties of emulsion systems stabilized by different mixing ratios of 1.0% TP-CP. (**A**) The average droplets size of emulsion systems over 21 days, (**B**) the ζ-potential of emulsion systems. Different letters in the same chart represent significant differences between different treatments (*p* < 0.05).

**Figure 6 foods-13-01545-f006:**
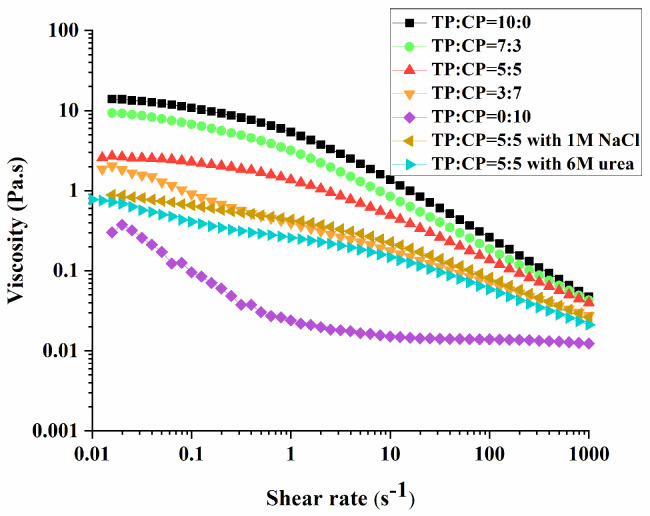
Apparent viscosity of 1.0% TP-CP solutions and 1.0% TP-CP (5:5) with NaCl or urea.

**Figure 7 foods-13-01545-f007:**
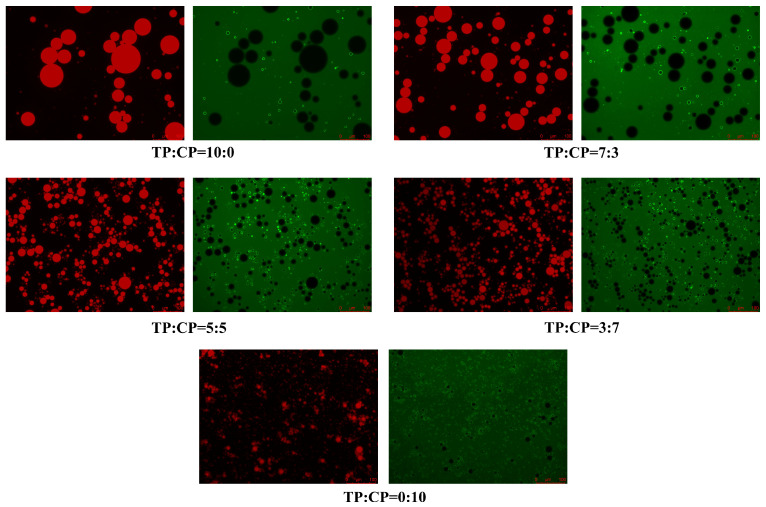
The images of the emulsions stabilized by different mixing ratios of 1.0% TP-CP. The polysaccharide appeared green after dyeing with FITC, while the oil droplets appeared red after dyeing with Nile Red.

**Figure 8 foods-13-01545-f008:**
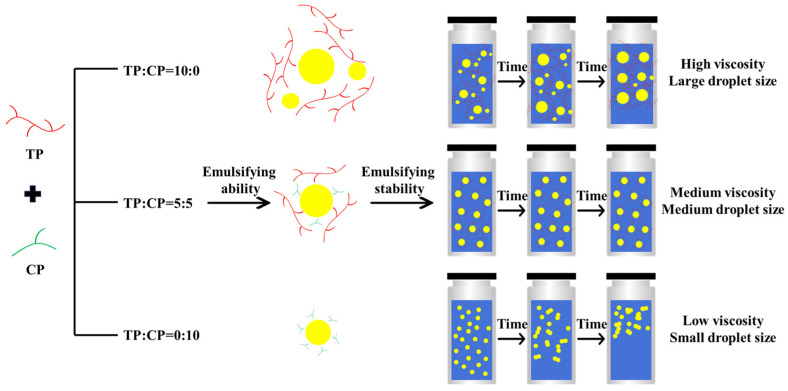
The possible mechanism of emulsion ability and stability of TP and CP.

**Table 1 foods-13-01545-t001:** The physicochemical properties of the two polysaccharides.

	TP	CP
Neutral sugar (%, *w*/*w*)	82.14 ± 0.77	28.56 ± 2.72
Uronic acid (%, *w*/*w*)	11.59 ± 0.32	68.70 ± 0.36
Protein (%, *w*/*w*)	2.03 ± 0.59	2.60 ± 0.52
Relative weight-average molecular weight (Da)	2.72 × 10^6^	1.28 × 10^6^
Monosaccharide composition (%, *w*/*w*)	Man:Fuc:Xyl:GlcA:Glc = 47:18:18:13:4	GalA:Gal:Rha:Glc = 76:12:8:4
DM (%)	-	69.16
DA (%)	-	2.01

Fuc: fucose, Rha: rhamnose, Gal: galactose, Glc: glucose, Xyl: xylose, Man: mannose, GalA: galacturonic acid, GlcA: glucuronic acid; “-” represents that the degree of methyl-esterification (DM) and acetylation (DA) of TP were not determined.

## Data Availability

The original contributions presented in the study are included in the article/Appendix A; further inquiries can be directed to the corresponding author.

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
