# Peer review of "The Emulsification and Stabilization Mechanism of an Oil-in-Water Emulsion Constructed from Tremella Polysaccharide and Citrus Pectin"

_foods, 2024, doi:10.3390/foods13101545_

Round 1

Reviewer 1 Report

Comments and Suggestions for Authors

This work has combined tremella polysaccharide and citrus pectin for stabilization of emulsions. The systems are characterized by various techniques, and the manuscript is well written too. The only question is how to confirm the the interactions of the two polysaccharides are hydrogen bonding and electrostatic interactions?

Author Response

Detailed Response to Reviewers’ Comments

Thanks for your constructive suggestions on our manuscript entitled “Emulsification and stabilization mechanism of oil–in–water emulsion constructed by tremella polysaccharide and citrus pectin” (ID: foods-2980332) to be published in Foods. We have carefully revised our manuscript according to your advice, addressing each point and marking the changes in the manuscript with red text. The response to comments and the manuscript have been uploaded as a Word file.

Reviewer 2 Report

Comments and Suggestions for Authors

Nice work but there are still two issues that need to be improved. First is English language that need to be checked by some software in order to remove grammatical and typo errors. Second is connected to some problems with Figures. There are also some suggestions about few explanation that author may or may not include. It is given in PDF document, but most of grammatical and typo errors are not pointed.

Comments on the Quality of English Language

Grammatical and typo error check is needed.

Author Response

(The authors gave the same response as above.)

Reviewer 3 Report

Comments and Suggestions for Authors

The manuscript describes a study of the properties of emulsions stabilised by polysaccharide mixtures. The authors conclude that the pectin conferred emulsifying activity to the system whilst the tremella polysaccharide conferred long term stability. The manuscript is clearly written, with a few minor grammatical errors. The data is clearly presented.  Although the technical aspects of the study are quite simple, the authors have made some attempt to investigate the mechanisms. There are however some issues with interpretation which could be improved with additional control experiments or further discussion, as described in my comments below.

The authors talk about a complex between TP and CP. How was the complex formed? What is the mechanism of interaction? Or did it occur spontaneously? Please provide some evidence that a complex is formed, otherwise it is just a mixture of the 2 polysaccharides.

Line 127 "11000 rpm for 120. The" -  no units - should it be 120 s. ?

Lines 235-238 Fig 3B. I agree the 3:7 sample shows clear signs of flocculation driven either by depletion or bridging. However the 10:0 and 0:10, the behaviour is not similar. The samples show no signs of flocculation, the progression of creaming is gradual with no sharp boundaries, indicating normal Stokesian creaming behaviour. However, the baseline of the 10:0 sample has reduced, suggesting some flocculation reducing the number of scattering centres, as evidenced by the scattering results Fig 6A, but the viscosity of the solution is high enough to slow down the mass creaming normally associated with this behaviour, as observed in 3:7. Please revise the discussion slightly to add this information.

Fig. 1S description "The morphologies of emulsion systems stabilized by 1.0% TP-CP at different concentrations." doesn't make sense. Should it be "stabilized by 5:5 TP-CP "

Fig. 6B. Did the authors check the zeta potential of the polysaccharides alone. At these molecular weights, it is likely that the polymers themselves will be detected by the zeta sizer. The authors should discuss why the TP shows a higher zeta potential than CP, as CP is supposed to be anionic.

Page 11 - section 3.5.4 - Yes, it is indeed speculation that TP and CP form a complex, but the authors should provide some evidence to support this, such as FTIR or HPGPC. Or add the side of the argument that these effects could also be observed with mixtures of non-interacting polymers each providing its own functionality to the emulsion.

Comments on the Quality of English Language

The quality of the English is pretty good. Just some minor typos and grammatical errors which can be picked up during editing

Author Response

(The authors gave the same response as above.)
